# Load Identification for the More Electric Aircraft Distribution System Based on Intelligent Algorithm

**Juan Yang** [1] , **Xingwang Bao** [2] **and Zhangang Yang** [2,*]

1   Engineering Techniques Training Center, Civil Aviation University of China, Tianjin 300300, China;
    haishi_yj11@126.com
2   College of Electronic Information and Automation, Civil Aviation University of China, Tianjin 300300, China;
    bao_xingwang@163.com
*   Correspondence: yangcauc@163.com

**Abstract:** Accurate identification of electrical load working status can provide information support to the remote electrical distribution system (EDS) of more electric aircraft (MEA), which could use it to realize redundant switching and protection. This paper presents a method to automatically identify the load status on the remote power distribution unit (RPDU) of MEA by using an intelligent algorithm. The experimental platform is built in an aircraft Electrical Power System (EPS) distribution large-scale test cabin. Four pieces of typical aviation equipment are installed in the test cabin and powered from RPDU. Voltage and current values under 15 working combinations on the RPDU are measured to extract the steady-state V-I trajectory. In total, 750 group samples were collected in the feature parameter database. A generalized regression neural network (GRNN) identification model was established, and the smoothing factor was calculated by using a conventional cross-validation method to train and reach an optimal value. However, the identification results are not ideal. In order to improve the accuracy, the parameter of GRNN was optimized by genetic algorithms. The proposed model shows great performance as accuracy of all 15 classifications reached 100%. The proposed model has advantages of flexible network structure, high fault tolerance, and robustness. It can realize global approximation optimization, avoid local optimization, effectively improve GRNN fitting accuracy, improve model generalization ability, and reduce model training calculation.

**Keywords:** load identification; aircraft power systems; more electric aircraft; generalized regression neural networks; genetic optimization





## 1. Introduction

With the development of more electric aircraft (MEA) and all electric aircraft (AEA), the type and quantity of electrical equipment on-board are increasing rapidly [1], and the power grid structure has also become more complex [2]. The problem of power distribution will arise [3]. At present, the aircraft electrical power system (EPS) distribution management mainly depends on the bus power control unit (BPCU), generator control unit (GCU), and APU generator control unit (AGCU) to detect the output of voltage, current, and frequency at the generator output monitoring points [4], as shown in Figure 1. The power control in Figure 1 only monitors the change in the output parameters of the generators and cuts off the corresponding contactor switching power supply logic when the parameters are abnormal, so as to keep the EPS supply normal. This control method does not pay attention to the working state of buses and loads in EPS, so it cannot accurately grasp the health state of the power grid to enable more scientific control.

Load identification could give more information regarding the load composition. It has the advantage that it is run online and considers the actual operating conditions (temperature, actual loading, etc.). It is more precise than an offline characterization [3].

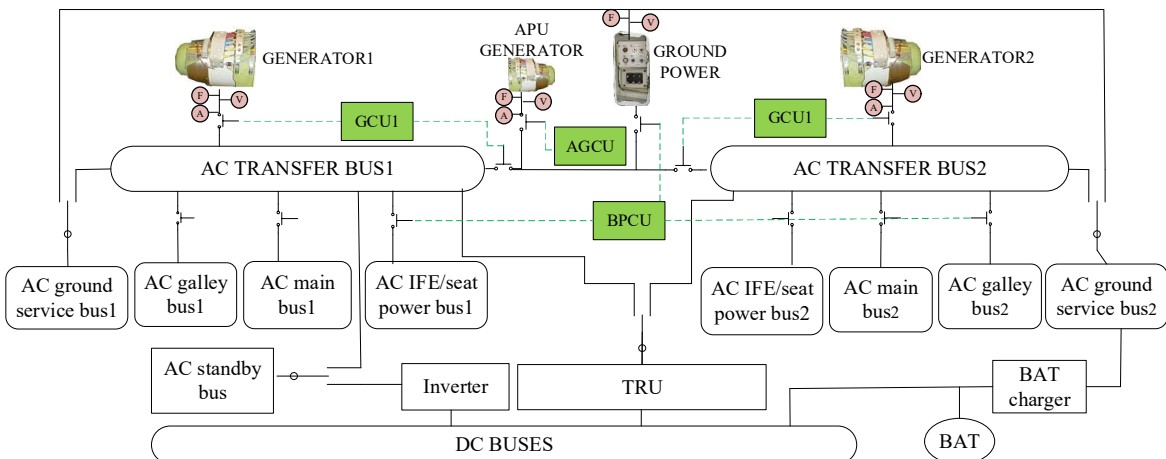

**Figure 1.** Typical aircraft power grid distribution structure.

In recent years, research on load identification mainly focuses on household electrical equipment and ship power systems [5,6]. Additionally, the intelligent identification technology is the most popular [7]. The process of power grid load identification can usually be completed by intelligent classification method [8]. The key technologies of intelligent identification mainly focus on load signature and classification algorithms [7,9]. Usually, the characteristic parameters of electric signals such as power, current waveform, current harmonics, and transient power spectrum are selected as the load signature [10,11]. Reference [12] selected steady-state active power and reactive power of the system to compare with the known power database to realize load decomposition. However, a lot of electrical equipment has similar active and reactive characteristics and it is hard to identify when facing the equipment in variable speed drive state. In [13,14], steady-state current waveform and current harmonics are taken as the load signature, but there are error-prone problems for similar equipment. In [15], transient power or current during load switching are used as the load signature, which has high identification accuracy, but requires high data acquisition and requires a large amount of calculation. Through comparative analysis, it is proved that devices with different working principles (such as resistance type, motor driven type, or power electronic type) have their own unique V-I trajectory images, which have high recognition. Therefore, choosing voltage–current (V-I) trajectory as the load signature has more advantages [16,17].

Meanwhile, machine learning algorithms, such as artificial neural network (ANN), deep learning, support vector machine (SVM), and K-nearest neighbor, are widely used in recognition technology [18]. However, both ANN and deep learning methods need a large amount of sample data to ensure their generalization ability [19,20]. In addition, there are too many algorithm parameters to consider and the amount of calculation is large. SVM has high recognition accuracy for single state equipment [21] and is not good at handling multi-state or continuously changing loads. The k-nearest neighbor method is mostly used to solve clustering problems, which is very sensitive to the selection of parameters, and each classification needs to calculate the distance between the unknown data and all training samples, which requires a large amount of calculation [22]. After comparison, the generalized regression neural network (GRNN) has a significant nonlinear mapping ability and strong approximation ability. However, this algorithm easily falls into the local optimum in practical calculations [23].

In order to provide load working state supports to bus load distribution management of MEA EPS, in this paper, a novel intelligent load identification technology is adopted. Voltage–current (V-I) trajectory is selected as the load signature. Four typical types of airborne equipment (transformer rectifier unit, battery charger, wing fuel boost pump, and anti-collision light) were loaded into the experimental cabin, and the V-I trajectory samples of all possible working combinations were collected in the experiment. The GRNN model

is obtained by training the sample data. Through the analysis of the identification results, it is known that the smoothing factor of the GRNN algorithm has a great influence on the accuracy of the model. In order to overcome the defects of the GRNN algorithm and speed up the calculation speed and identification accuracy, a genetic algorithm is proposed to optimize the parameters of the GRNN model. Through the test sample, the genetic optimized GRNN identification method can accurately realize the task of identifying the working state of the load.

## 2. MEA Distribution Model and Load Working State Analysis

Remote distribution is an advanced distribution technology used on MEA [24]. Analyzing the EPS distribution structure of MEA, and further analyzing the working states of all kinds of equipment on the bus bars, is helpful for establishing algorithm research of load identification.

### 2.1. MEA Distribution Model

As depicted in Figure 2, although a lot of high-power electrical equipment is directly powered by primary buses and secondary buses, about 85% of the electrical equipment on board is low-power equipment, which is powered through RPDU [25,26]. Power management includes bus bar control and RPDU control. The bus control module monitors and controls the power consumption of AC and DC bus bars at all levels. The RPDU control module monitors the working status of the electrical equipment on the RPDU in real time and sends control instructions according to the actual work needs.

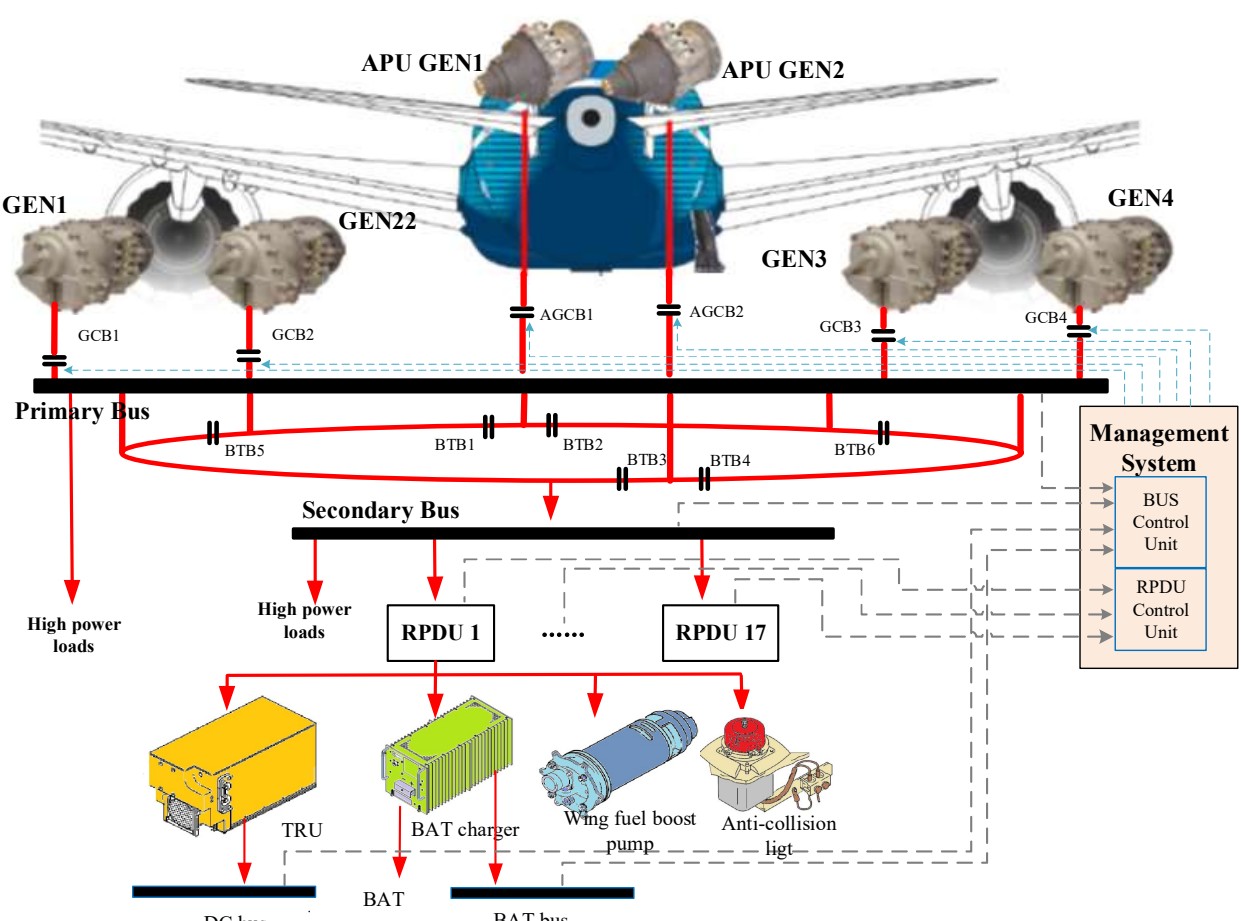

**Figure 2.** Schematic of aircraft distributed power supply grid.

### 2.2. Load Multi-Modal Working Combination

Taking the RPDU1 in Figure 2 as the object, the working states of the transformer rectifier unit (TRU), battery (BAT) charger, wing fuel boost pump, and anti-collision light are analyzed. There are nine working states of the aircraft in one cycle, which are ground maintenance, loading and preparation, startup and preheating, taxiing, takeoff and climb, cruise, night flight or de-icing, landing, and emergency state. Under different working conditions, the working state of the onboard electrical equipment changes accordingly. When the aircraft is on the ground, the TRU and anti-collision lamp are in working state during ground maintenance, loading, and preparation. In the startup and preheating period, the auxiliary power unit is started by the onboard battery, so the battery charger is in the voltage transformation and rectification mode. In the full cycle condition, the aircraft is supplied with fuel from the central tank, and the wing fuel boost pump is usually in standby state. During the failure of the central fuel boost pump, the wing boost fuel pump will start to work. When cruising and landing, the battery charger is switched to charging mode to charge the battery. If in night navigation or de-icing, the demand for power capacity increases and the power of common equipment on board will be reduced as required. The emergency state touches the deadline of airworthiness configuration, and the TRU needs to work at full load to ensure the normal power supply of the DC bus. In order to fully reflect various possible load states of the bus, Table 1 lists all possible working combination classifications of electrical equipment on RPDU1 [27,28].

**Table 1.** Load working combination classification.

| | Name | TRU | BAT Charger | Wing Fuel Boost Pump | Anti-Collision Light |
|---|---|---|---|---|---|
| Categories | 1 | 1 | | | |
| | 2 | | 1 | | |
| | 3 | | | 1 | |
| | 4 | | | | 1 |
| | 5 | 1 | 1 | | |
| | 6 | 1 | | 1 | |
| | 7 | 1 | | | 1 |
| | 8 | | 1 | 1 | |
| | 9 | | 1 | | 1 |
| | 10 | | | 1 | 1 |
| | 11 | 1 | 1 | 1 | |
| | 12 | 1 | 1 | | 1 |
| | 13 | 1 | | 1 | 1 |
| | 14 | | 1 | 1 | 1 |
| | 15 | 1 | 1 | 1 | 1 |

## 3. Experimental Setup

An experimental platform is built in an aircraft EPS distribution large-scale test cabin to collect the voltage and current data of all working states of a certain RPDU on the EPS distribution system. Through data preprocessing, V-I trajectory samples are formed, some of which are used as training samples for intelligent identification algorithms, and the other as test samples.

### 3.1. Experimental Platform

The experimental platform is built in an aircraft EPS distribution large-scale test cabin, as depicted in Figure 3. The power supply source is led out from the RPDU in the cabin section. The four pieces of airborne equipment, TRU, BAT charger, wing fuel boost pump, and anti-collision light are powered from the power grid. The switch control panel controls the switching of 15 working combinations according to Table 1. The current transformer and voltage sensor measure the steady-state electrical parameters on the line, and the multi-channel data recorder records the sample data.

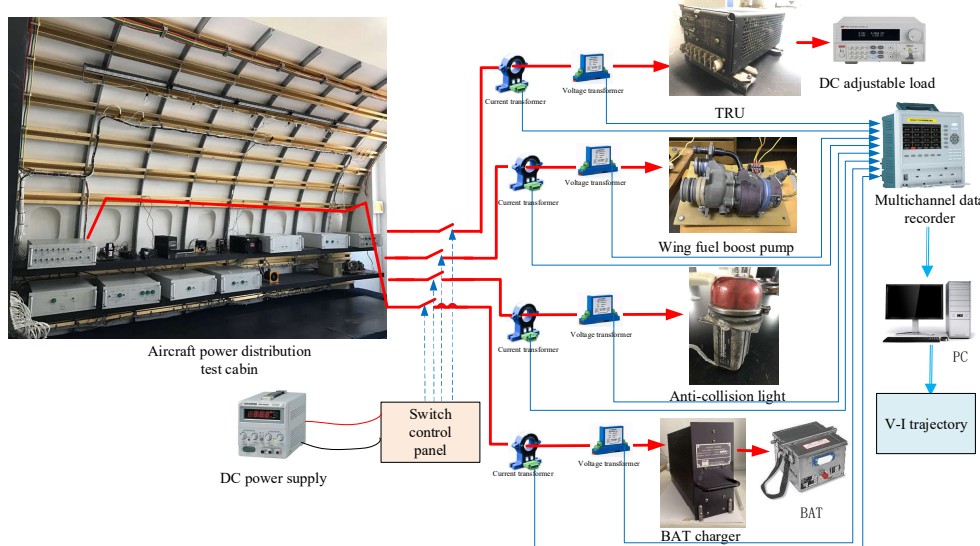

**Figure 3.** Schematic of the experimental platform in a large test-aircraft cabin.

### 3.2. Sample Collection and Processing

The voltage and current data were collected in a complete cycle after the load switching operation was stable under each working combination. Since the measured signal contains a certain amount of system operation status information and the signal-to-noise ratio is low, it is necessary to reduce the noise of the original signal. The data with noise was processed by fast Fourier transform and windowing on the MATLAB platform, and the data was smoothed by moving average method.

After smoothing, the noise interference was effectively eliminated, and the processed current and voltage data were fitted to obtain the V-I trajectory characteristic curve, as shown in Figure 4. The steady-state voltage and current waveforms basically include all the steady-state electrical characteristics of the power load. It can be seen from Figure 4 that the V-I trajectory converts the characteristic parameters of two independent values of voltage and current into the parameters combining the relative relationship between voltage and current as the identification mark. The constructed steady-state V-I trajectory can completely display typical steady-state characteristics of the load. The V-I trajectory reflected by the bus supplying power to a single device and multiple devices will have different characteristics. Additionally, the change in equipment switching will produce obvious changes.

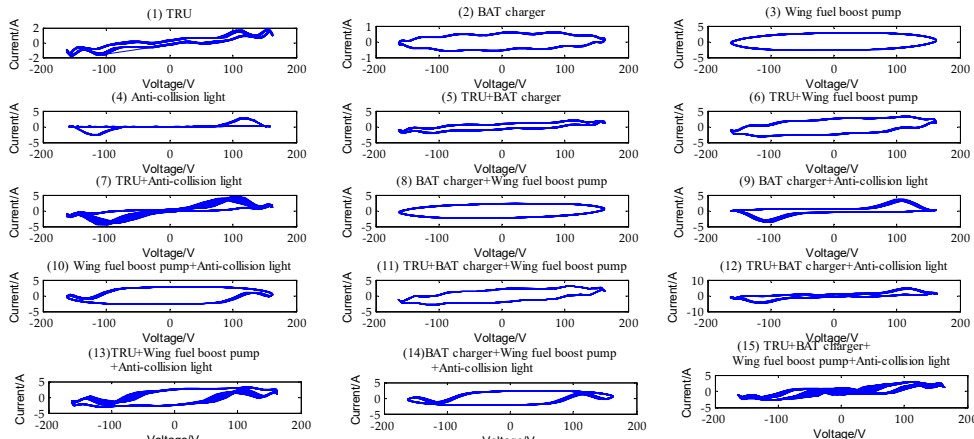

**Figure 4.** V-I trajectory of load working combination on RPDU1.

## 4. Algorithm Analyses

Compared with the transfer type neural network algorithm, GRNN does not need to set the network connection weight manually. It has a strong nonlinear fitting ability and flexible network structure. However, the GRNN network construction process is simple, and the smoothing factor has a great influence on the network performance. The appropriate smoothing factor value not only takes all the training samples into account according to the distance between different training sample points and the test input sample, but also avoids the phenomenon of over-learning.

The genetic algorithm has group search characteristics, is not easy to limit to local optimum and has the advantages of parallel computing ability. Using the global optimization ability of the genetic algorithm to optimize the smoothing factor of GRNN can effectively improve the model accuracy.

### 4.1. GRNN Algorithm

GRNN is a type of radial basis function neural network. It combines the typical radial basis function neural network with the numerical statistics theory and is composed of an input layer, mode layer, summation layer, and output layer [7].

The number of neurons in the input layer is the same as the dimension of the input vector in the mode layer, so the input variables are directly transferred to the mode layer. Each neuron of the mode layer corresponds to different samples. The transfer function of the mode layer neuron is described as:

$$P_i = e^{\left[ -\frac{(X-X_i)^T(X-X_i)}{2\sigma^2} \right]}, \ i = 1, 2 \ldots \ldots n \tag{1}$$

where $P_i$ is the output of the $i$th neuron in the mode layer, $X_i$ represents the training samples corresponding to the $i$th neuron, and $\sigma$ is the smoothing factor.

The model consists of two classes of confluent neurons. Among them, the denominator neuron summation is the arithmetic summation of the outputs of neurons in all mode layers. It can be described with the following formula:

$$S_D = \sum_{i=1}^{n} P_i = \sum_{i=1}^{n} e^{\left[ -\frac{(X-X_i)^T(X-X_i)}{2\sigma^2} \right]}, \ j = 1, 2 \ldots \ldots k \tag{2}$$

The connection weight between the mode layer and summation layer is 1.

Molecular neuron summation sets the connection weight between the $i$th neuron in the summation layer and the $j$th neuron in the mode layer as the $i$th element in the $j$th input sample, and performs weighted summation on all mode layer neurons, described as the following formula:

$$S_{N_j} = \sum_{i=1}^{n} Y_i P_i = \sum_{i=1}^{n} y_{ij} e^{\left[ -\frac{(X-X_i)^T(X-X_i)}{2\sigma^2} \right]}, \ j = 1, 2 \ldots \ldots k \tag{3}$$

The output layer divides the above two types of neuron outputs of the summation layer to obtain the estimated output value of the neuron, so that:

$$y_j = \frac{S_{N_j}}{S_D}, j = 1, 2 \ldots \ldots k \tag{4}$$

Output $y_j$ is the $j$th element of the estimation result $\hat{Y}(X)$. The prediction output of the GRNN neural network is:

$$\hat{Y}(X) = \frac{\sum_{i=1}^{n} Y_i e^{\left[ -\frac{(X-X_i)^T(X-X_i)}{2\sigma^2} \right]}}{\sum_{i=1}^{n} e^{\left[ -\frac{(X-X_i)^T(X-X_i)}{2\sigma^2} \right]}} \tag{5}$$

Assuming the joint probability density function of input variable $x$ and output variable $y$ is $f(x, y)$. Density function $\hat{f}(X, y)$ can be estimated through the sample data $X_i$ and $Y_i$ according to:

$$\hat{f}(X, y) = \frac{\sum_{i=1}^{n} e^{\left[-\frac{(X-X_i)^T(X-X_i)}{2\sigma^2}\right]} e^{-\frac{(X-Y_i)^2}{2\sigma^2}}}{n(2\pi)^{\frac{t+1}{2}} \sigma^{(t+1)}} \tag{6}$$

where $t$ is the dimension of $X$.

*4.2. GRNN Algorithm Accuracy Analyses*

Precision (PRE) and recall (REC) are suitable performance measures for classification models. Precision and recall are contradictory metrics. Generally speaking, when the precision rate is high, the recall rate tends to be low, and when the recall rate is high, the precision rate tends to be low. Therefore, harmonic average $F_1$ is taken to comprehensively consider the performance metrics of precision and recall. The precision, recall, and the harmonic average $F_1$ are used as the result evaluation parameters.

4.2.1. Evaluation Methodology

The precision of the identification result is:

$$PRE = \frac{TP}{TP + FP} \tag{7}$$

where $TP$ is the number of correctly identified samples and $FP$ is the number of incorrectly identified samples.

The recall of the identification result is:

$$REC = \frac{FP}{TP + FP} \tag{8}$$

The harmonic average $F_1$ of the identification result is:

$$F_1 = \frac{2 \times PRE \times REC}{PRE + REC} \tag{9}$$

4.2.2. Smoothing Factor Influence of GRNN

Experimental test and verification were adopted to reveal the influence of the smoothing factor σ on the accuracy of the GRNN model. Through model calculation, the changes in PRE and REC corresponding to different smoothing factors in the range of 0~1 in GRNN model are shown in Table 2.

**Table 2.** Influence of smoothing factor on precision and recall.

| | | 1 | 2 | 3 | 4 | 5 | 6 | 7 | 8 | 9 | 10 | 11 | 12 | 13 | 14 | 15 |
|---|---|---|---|---|---|---|---|---|---|---|---|---|---|---|---|---|
| | | | | | | | | PRE% | | | | | | | | |
| | 0.1 | 100 | 81.25 | 9.09 | 100 | 100 | 100 | 70 | 100 | 100 | 100 | 100 | 100 | 100 | 100 | 100 |
| | 0.2 | 100 | 76.65 | 9.33 | 85.71 | 100 | 100 | 50 | 100 | 100 | 100 | 100 | 100 | 37.9 | 68.42 | 100 |
| | 0.3 | 100 | 46.14 | 0 | 45 | 66.67 | 100 | 52.38 | 100 | 100 | 100 | 100 | 100 | 92.31 | 100 | 100 |
| | 0.4 | 100 | 64.70 | 0 | 44.44 | 100 | 100 | 48 | 100 | 100 | 100 | 100 | 100 | 100 | 100 | 100 |
| σ | 0.5 | 100 | 70 | 0 | 32.14 | 100 | 100 | 61.5 | 100 | 100 | 100 | 100 | 100 | 100 | 100 | 100 |
| | 0.6 | 100 | 60.87 | 0 | 46.15 | 100 | 100 | 53.84 | 100 | 100 | 100 | 100 | 100 | 100 | 100 | 100 |
| | 0.7 | 100 | 68.75 | 0 | 39.39 | 22.22 | 100 | 35.7 | 0 | 100 | 100 | 52.94 | 42.85 | 33.33 | 0 | 100 |
| | 0.8 | 100 | 92.30 | 0 | 83.33 | 26.31 | 100 | 53.84 | 0 | 100 | 100 | 62.5 | 55 | 53.33 | 0 | 100 |
| | 0.9 | 100 | 75 | 0 | 84.6 | 53.3 | 92.86 | 66.67 | 0 | 78.57 | 100 | 71.43 | 37.5 | 30 | 0 | 100 |
| | 1.0 | 100 | 75 | 0 | 100 | 61.54 | 58.33 | 47.06 | 0 | 66.67 | 100 | 52.94 | 33.33 | 20 | 0 | 100 |
| | | 1 | 2 | 3 | 4 | 5 | 6 | 7 | REC% 8 | 9 | 10 | 11 | 12 | 13 | 14 | 15 |
| | 0.1 | 65 | 100 | 7.65 | 100 | 100 | 100 | 70.6 | 88.54 | 100 | 100 | 100 | 100 | 100 | 100 | 100 |
| | 0.2 | 70 | 100 | 9.33 | 100 | 85.7 | 100 | 100 | 68.75 | 100 | 100 | 100 | 100 | 100 | 100 | 100 |
| | 0.3 | 46.15 | 100 | 0 | 100 | 57.14 | 100 | 100 | 44.44 | 100 | 100 | 100 | 100 | 100 | 90 | 100 |
| | 0.4 | 45.45 | 100 | 0 | 100 | 100 | 100 | 100 | 48.75 | 100 | 100 | 100 | 100 | 100 | 100 | 100 |
| σ | 0.5 | 50 | 100 | 0 | 100 | 38.89 | 100 | 100 | 58.3 | 100 | 100 | 100 | 100 | 100 | 100 | 100 |
| | 0.6 | 47.08 | 100 | 0 | 100 | 70 | 100 | 100 | 40 | 100 | 100 | 100 | 100 | 88.89 | 100 | 100 |
| | 0.7 | 44.44 | 100 | 0 | 100 | 13.3 | 83.3 | 100 | 0 | 100 | 100 | 100 | 27.25 | 55.56 | 0 | 100 |
| | 0.8 | 50 | 100 | 0 | 100 | 71.43 | 90 | 100 | 0 | 100 | 100 | 100 | 25 | 61.54 | 0 | 100 |
| | 0.9 | 40 | 100 | 0 | 100 | 100 | 80 | 92.3 | 0 | 100 | 70 | 100 | 20 | 25 | 0 | 100 |
| | 1.0 | 20 | 100 | 0 | 100 | 100 | 63.63 | 72.72 | 0 | 100 | 45.45 | 100 | 12.5 | 20 | 0 | 100 |

In Table 2, the smaller the smoothing factor is, the more accurate the approximation of the function will be. The larger the smoothing factor is, the smoother the function fitting will be. However, the approximation error of 15 working status categories will become larger. The results indicate that the values of PRE and REC of identification results decrease with the increase in smoothing factor. When the smoothing factor value is close to 1, the identification precision of categories 3, 8, 12, 13, and 14 tends to 0, and the recall of categories 1, 3, 6, 8, 12, 13, and 14 tends to 0, which indicates the failure of the identification task. The value of the smoothing factor σ of GRNN has a great influence on the approximation results of the model.

### 4.3. Genetic Algorithm Analyses

The genetic algorithm is a parallel random search optimization method proposed by Professor Holland of Michigan University in the United States to simulate the genetic mechanism of nature and biological evolution theory. The genetic algorithm introduces the biological evolution principle of "survival of the fittest" into the coding tandem population formed by the optimization parameters. According to the selected fitness function, individuals are screened through selection, crossover, and mutation to retain the individuals with good fitness and eliminate the individuals with poor fitness. The new population not only inherits the information of the previous generation, but also is superior to the previous generation. After several generations of circulation, the conditions are met. The design of genetic algorithm includes initialization population construction, fitness function calculation, genetic operator calculation, and control parameter setting [23]. The flow of the genetic algorithm is depicted as Figure 5.

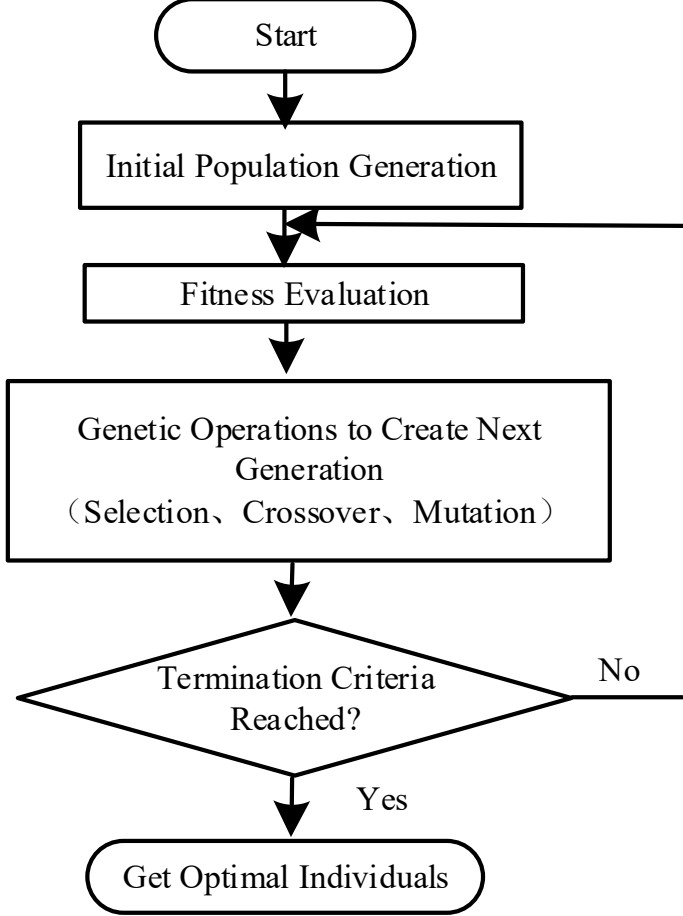

**Figure 5.** The flowchart of the genetic algorithm.

The genetic algorithm is proposed to optimize the initial value of smoothing factor σ of the GRNN model. The initial population is generated randomly, and a certain scale of smoothing factor initial population is obtained using the real number coding method.

The square sum of the error is selected between the model output value and the training real value as the fitness function, which is defined as:

$$f_i = \sum_{i=1}^{n} (Y_i - \overset{\wedge}{Y_i})^2 \tag{10}$$

where $f_i$ is the fitness value of the individual $i$, $Y_i$ and $\overset{\wedge}{Y_i}$ represent the real value and predicted value of the training samples, and $n$ is the number of training samples.

Genetic operators include selection, crossover, and mutation. Selection is a process of reproduction. Excellent individuals are selected from the current population according to a certain probability, so that they have the opportunity to reproduce the next generation as parents. The roulette method is adopted, and the selection strategy is based on the fitness ratio. The selection probability of individuals is:

$$p_i = f_i / \sum_{j=1}^{N} f_j \tag{11}$$

where $p_i$ is selection probability of individual $i$, $N$ is the population size, and $f_j$ is the fitness value of individual $j$.

The individuals are encoded by real numbers, the crossover operation method is arithmetic crossover, and the simple single point crossover operation is used to randomly generate intersections.

The crossover equation is:

$$\begin{cases} a'_l = b \cdot a_l + (1 - b) \cdot a_n \\ a'_n = b \cdot a_n + (1 - b) \cdot a_l \end{cases}, b \in [0, 1] \tag{12}$$

where $a_l$ and $a_n$ are the individuals before crossover, $a'_l$ and $a'_n$ are the individuals after crossover, and $b$ represents random coefficients.

The $j$th gene in the $i$th individual variation is selected and the equation for variation is:

$$a'_p = \begin{cases} a_p + (a_p - a_{\max}) \cdot r \cdot (1 - \frac{g}{G_{\max}})^2, 1 \geq r > 0.5 \\ a_p + (a_{\min} - a_p) \cdot r \cdot (1 - \frac{g}{G_{\min}})^2, 0 < r \leq 0.5 \end{cases} \tag{13}$$

where $a_p$ is the individual before mutation, $a'_p$ is the individual after mutation, $a_{\min}$ and $a_{\max}$ are the lower limit and upper limit of the individual, $r$ is the random coefficient, $g$ is the current iteration number, and $G_{\max}$ is the maximum evolution algebra.

### 4.4. Genetic Algorithm Parametric Analysis

In the genetic algorithm, the population size affects the optimization ability of the algorithm. The differential value between the real category and identification category changing with the population size is depicted as Figure 6. The fitness change curve is depicted as Figure 7.

In Figure 6, as the population size increases from 5 to 30, the identification differential value converges to 0.

In Figure 7, when the population size is 5, the fitness of the first generation is 9.13. With the increase in evolutionary algebra, the minimum fitness reaches 1.03; the fitness curve was improved obviously with the increase in population size. When the population size was 30, the fitness of the first generation was 1.09 and the optimal value was quickly found.

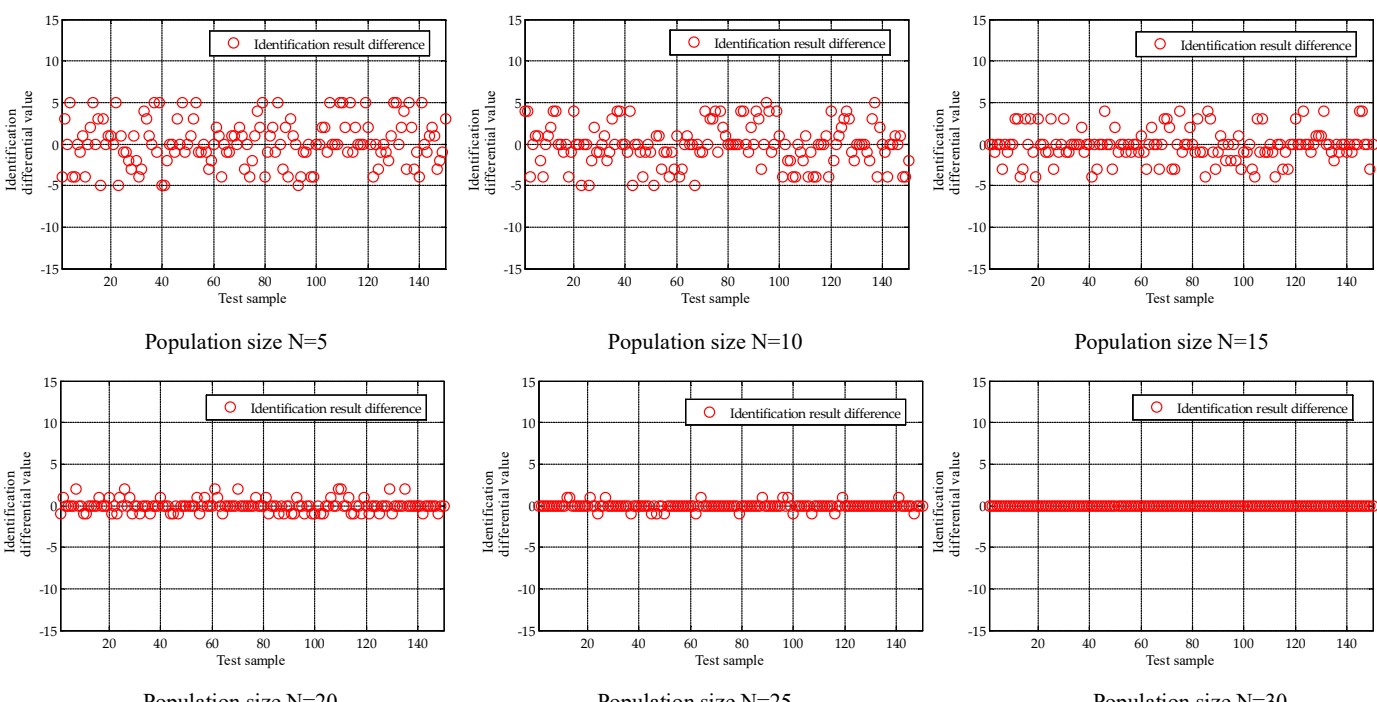

**Figure 6.** Influence of population size on identification error.

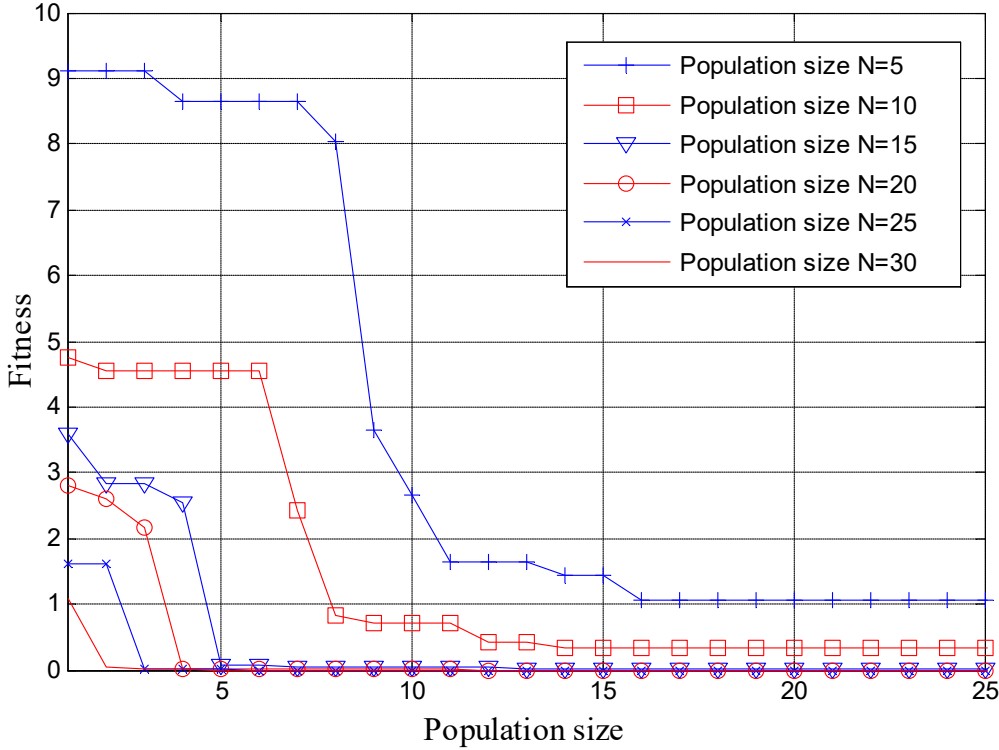

**Figure 7.** Effect of population size on fitness.

## 5. Experimental Results

Two models of cross-validation GRNN and genetic optimized GRNN are used to identify the load working state categories on RPDU1, and the identification results are analyzed and compared.

### 5.1. GRNN Model Identification Result and Analyses

The experimental platform collected 50 groups of voltage and current waveforms in 15 working conditions, forming a total of 750 groups of V-I trajectory samples. The GRNN algorithm was adopted to establish the recognition model, 600 of 750 groups of samples were randomly selected as model training samples, and the remaining 150 groups of samples were used for model verification. The conventional cross-validation method was used to obtain the optimal value of the GRNN smoothing factor. The comparison between the identification results after 4 cross verifications and the actual categories is shown in Figure 8a, and the recall, precision, and $F_1$ parameters of the recognition results are shown in Figure 8b.

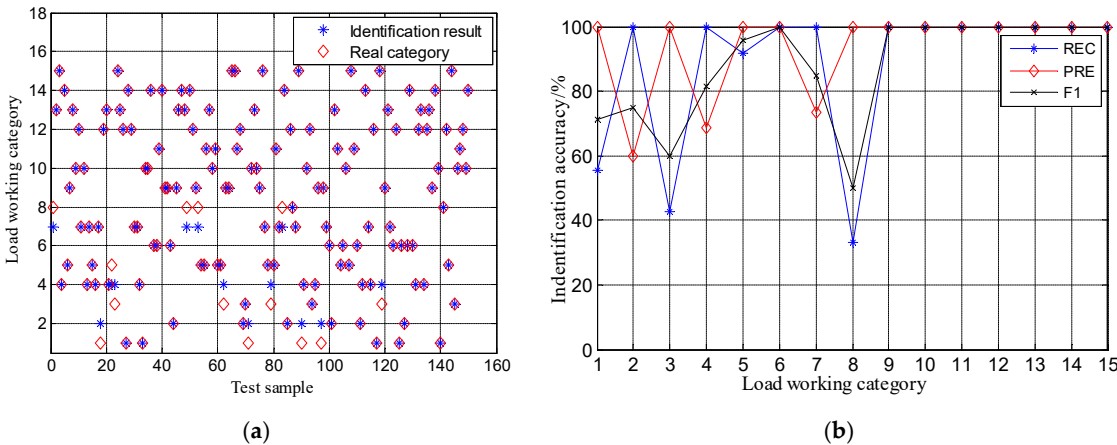

(**a**)　　　　　　　　　　　　　　　　　　　　　　　　(**b**)

**Figure 8.** GRNN model identification results and accuracy: (**a**) comparison between recognition results and actual categories; (**b**) identification accuracy of the GRNN model.

According to the analysis of Figure 8, there are 13 groups of sample-classification errors in 150 groups of test samples, mainly concentrated in categories 1, 2, 3, 4, 5, 7, and 8. The identification results of the remaining categories are all zero errors.

Considering that there are certain differences in each calculation of model classification, the GRNN algorithm identification results were counted 8 times, and 100% of the results were omitted. The categories with precision and recall less than 100% are shown in Table 3.

**Table 3.** Statistics of precision and recall less than 100%.

| | | 1st | 2nd | 3rd | 4th | 5th | 6th | 7th | 8th |
|---|---|---|---|---|---|---|---|---|---|
| | | **PRE/%** | | | | | | | |
| | | 1st | 2nd | 3rd | 4th | 5th | 6th | 7th | 8th |
| Category | 2 | 60 | 76.92 | 60.0 | 62.5 | 60 | 81.25 | 63.16 | 93.33 |
| | 4 | 68.755 | 66.67 | 58.82 | 60 | 62.5 | 57.15 | 66.67 | 73.33 |
| | 7 | 73.33 | 70.59 | 50 | 72.73 | 62.5 | 60 | 50 | 68.75 |
| | | **REC/%** | | | | | | | |
| | | 1st | 2nd | 3rd | 4th | 5th | 6th | 7th | 8th |
| Category | 1 | 55.56 | 66.67 | 42.86 | 53.85 | 57.15 | 75 | 41.67 | 90 |
| | 3 | 42.86 | 14.29 | 28.57 | 16.67 | 37.5 | 20 | 55.56 | 57.14 |
| | 5 | 91.67 | 90 | 84.62 | 87.5 | 87.5 | 77.78 | 85.71 | 85.71 |
| | 8 | 33.33 | 37.5 | 85.71 | 23.08 | 25 | 27.27 | 28.57 | 28.57 |

Considering synchronizing the V-I trajectory in Figure 4, identification results in Figure 8, and accuracy data in Table 2, it can be seen that the accuracy of No. 11, 12, 13, 14, and 15 categories always remain at 100%; that is, when more than 3 pieces of equipment work at the same time, the V-I trajectory features significantly and the identification accuracy is high. The identification results of working categories 6, 9, and 10 are also 100%. Based on the analysis in Figure 8, the V-I trajectory graphics of these three working states are highly

identifiable and are not easily confused with other V-I trajectories. The PRE of No. 1, 3, 5, and 8 working states is high, but the REC is low; the REC of No. 2, 4, and 7 working states is high and the PRE is low. According to Figure 4, comparing with the high-power wing fuel boost pump, the working load intensity of the BAT charger has little impact on the circuit, so it has little impact on the V-I trajectory. Based on this, it is difficult to distinguish the working state categories of No. 3 and No. 8. Similarly, the classification error rate of No. 1 and No. 5 is high. In addition, the pulse stroboscopic characteristic of the anti-collision light makes it easy to identify according to its V-I trajectory. The trajectory paths of No. 4 and No. 7 are similar and difficult to identify. The accuracy of 8 times identification results is not ideal to a certain extent, which shows that the generalization ability of the cross-validation method is limited.

### 5.2. Parameter Set of the Genetic Algorithm Model

The genetic algorithm was used to optimize the GRNN model to complete the identification task. The parameter settings of genetic algorithm are shown in Table 4.

**Table 4.** Parameter setting of the genetic algorithm.

| Parameter Name | Value |
| --- | --- |
| Population size | $N = 30$ |
| Generation of series | $G_{max} = 25$ |
| Crossover rate | 0.8 |
| Mutation rate | 0.1 |

### 5.3. Genetic Algorithm Optimized GRNN Model Identification Result and Analyses

Using 750 groups of experimental samples collected from the same group of experiments, the fitness change in the genetic optimization process is shown in Figure 9.

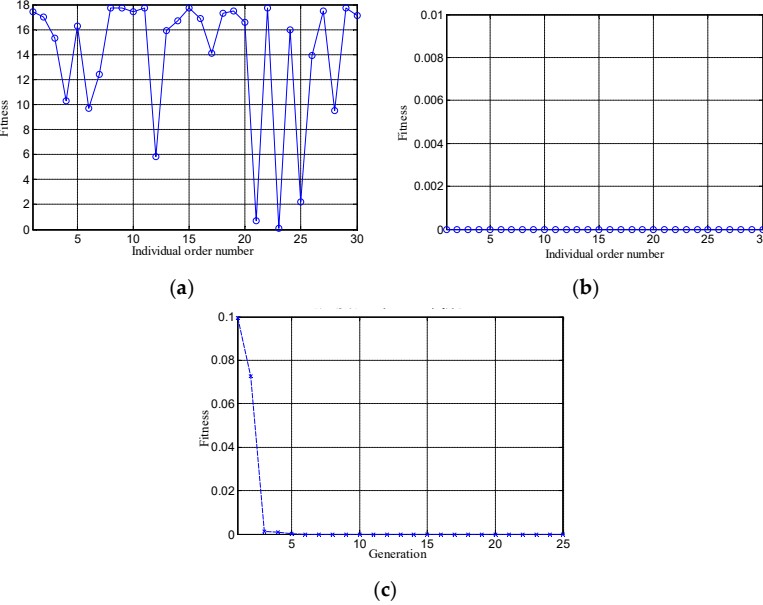

**Figure 9.** Fitness results of genetic algorithm optimized GRNN: (**a**) individual fitness of the first-generation population; (**b**) individual fitness of the final generation population; (**c**) fitness curve.

Figure 9a shows the individual fitness values of the first-generation population. Since the initial population is randomly generated, the fitness values obtained differ greatly, and the population fitness values are unstable, with a maximum value of 17.7; the fitness value of the 23rd individual in the population is 0.1, the value is the smallest, and the corresponding smoothing factor is 0.6. Figure 9b shows the fitness values of the last

generation of individuals. The fitness values of all individuals in the population reached the minimum value, and the trend is stable. The fitness curve changes in all generations are shown in Figure 9c. According to the analysis of Figure 9c, since the optimization parameter is only a real number, the fitness curve quickly converges, and the global optimization begins to be achieved in the 5th generation genetic population.

The result, error, and accuracy of the optimized GRNN algorithm are shown in Figure 10. The classification results are shown in Figure 10a, and the identification recall, precision, and $F_1$ parameters are shown in Figure 10b.

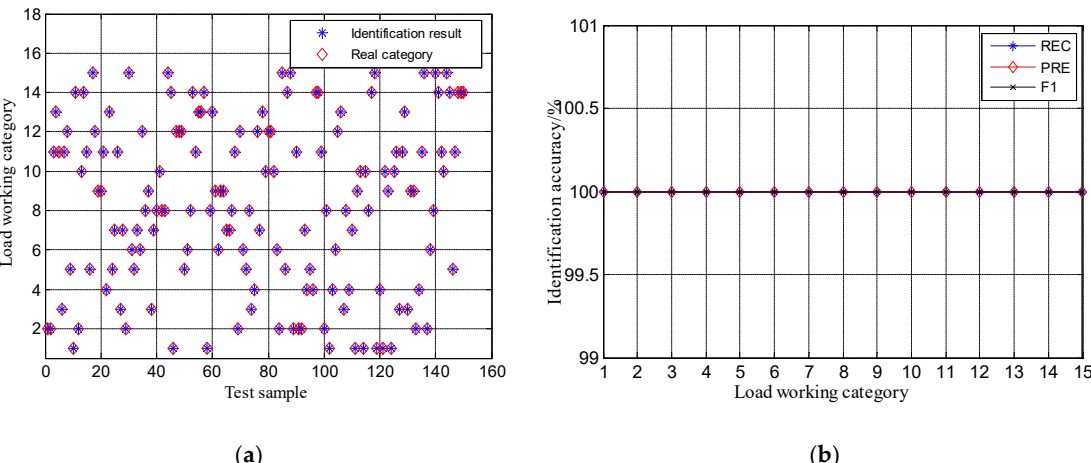

(**a**)　　　　　　　　　　　　　　　　　　　　　　　　(**b**)

**Figure 10.** Identification results and accuracy of genetic algorithm optimized GRNN: (**a**) comparison between recognition results and actual categories; (**b**) identification accuracy.

By analyzing Figure 10a,b, the identification results of all 150 verification samples of No. 1~15 categories are consistent with the actual categories. The recall, precision, and $F_1$ of identification of all categories reached 100%, which is significantly higher than the identification ability of the conventional cross-validation GRNN algorithm. The identification results lower than 100% in Table 2 were effectively solved.

Optimizing the GRNN model using a genetic algorithm has the advantage of flexible network structure, high fault tolerance, and robustness. It can realize global approximation optimization, avoid local optimization, effectively improve GRNN fitting accuracy, improve model generalization ability, and reduce model training calculations.

## 6. Conclusions

The application of an intelligent algorithm to the identification of electrical load working state of aircraft EPS not only has high precision and fast calculation speed, but also can provide more comprehensive and effective data support for system state detection and health management [29]. Considering the large amount of on-board electrical equipment and the problems of large amounts of calculation and difficult implementation of component level monitoring and management, it is more practical to adopt bus bar level power grid monitoring and control.

(1)　The V-I trajectory is used to characterize the steady-state electrical characteristics at the bus bar level, and can achieve certain effects as a single equipment identification feature or multiple equipment identification features.

(2)　When there are more than three pieces of electric equipment working on the bus bar at the same time, the V-I trajectory characteristics are obvious. The monitoring and identification accuracy is high, which can reach 100%.

(3)　Cross-validation is the most common method in GRNN model parameter optimization. However, it is proved by experiments that it is not suitable for bus-bar level load identification. While the difference between the V-I trajectories is small, the GRNN algorithm has the shortcomings of low precision and recall. For example, the

identification precision of both the transformer rectifier and anti-collision light are very low, up to 57.1% minimum, because their trajectories are very similar; when the difference between the V-I trajectory is obvious, the identification accuracy is also low when they work at the same time. For example, the identification recall rate of the wing fuel boost pump and battery charger working at the same time (category 8) and fuel boost pump working alone (category 3) is as low as 23.08% and 14.29%.

(4) The genetic optimization GRNN algorithm has fast learning speed and meets the requirements of fast response speed of aircraft system management. Selecting an appropriate population size can effectively improve the accuracy of the algorithm. It has a strong approximation ability and high identification accuracy, and the identification rate of V-I trajectory samples for 15 working combinations reaches 100%. This model is obviously superior to the GRNN model obtained by cross-validation.

This study was based on the normal working state of on-board loads. When the load fails to operate or work fails due to the influence of the electrical power quality, the load working state combination category will increase, and the V-I trajectories are all different. The proposed identification model in this paper will need to be further improved to cover all possibilities.

The genetic optimization GRNN algorithm monitors the steady-state V-I trajectory and identifies the power consumption status on the RPDU. However, the structure of the aircraft EPS distribution system is complex; for example, there are more than a dozen RPDUs. In addition, the flight conditions are also very complicated. The algorithm will be very complex for load identification under the working state of multi-field coupling. These aspects will be considered in the next stage of research.

**Author Contributions:** Conceptualization, J.Y. and Z.Y.; methodology, J.Y.; software, J.Y.; validation, J.Y. and X.B.; formal analysis, J.Y.; investigation, X.B.; resources, J.Y.; data curation, J.Y. and X.B.; writing—original draft preparation, J.Y.; writing—review and editing, Z.Y. All authors have read and agreed to the published version of the manuscript.

**Funding:** This research was funded by the Key Support Project of Civil Aviation Joint Fund of National Natural Science Foundation of China (NSFC), grant number U2033204. This research was also funded by the Fundamental Research Funds for the Central Universities, grant number 3122020029.

**Institutional Review Board Statement:** Not applicable.

**Informed Consent Statement:** Not applicable.

**Data Availability Statement:** The data presented in this study are available on request from the corresponding author.

**Acknowledgments:** The authors would like to acknowledge all team-mates.

**Conflicts of Interest:** The authors declare no conflict of interest.

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
