# Peer review of "Load Identification for the More Electric Aircraft Distribution System Based on Intelligent Algorithm"

_aerospace, doi:10.3390/aerospace9070350_

Round 1

Reviewer 1 Report

In this paper, the authors proposed a load identification based on genetic algorithm for the electrical distribution system of electric aircrafts. Even though it shows consistent development, some aspects to review and improve are:

1. It is recommended to review and better structure the title since it is not appropriate to use acronyms.

2. Check the definition of the acronym MEA on line 13, it should be: more electric aircraft (MEA). Besides in line 16: electrical power system (EPS).

3. I recommend including an introductory paragraph in Sections 2, 3, 4, 4.2, and 5, indicating the topics described in these sections. This allows having a common thread at the time of reading.

4. Present the flowchart of the genetic algorithm used.

5. In equation 10 use suitable parentheses to cover each fraction.

6. Please justify the parameter selection of the genetic algorithm.

7. Considering the stochastic behavior of the genetic algorithm, it is suggested to add a statistical analysis to strengthen the results regarding the methodology of parametric and non-parametric tests implemented to determine the comparison of cases (ANOVA, Kruskal Wallis, Bonferroni…). If the authors consider that these statistical tests are not feasible, it must be justified in the discussion section.

8. It is important to include a discussion section where the effectiveness, limitations, and aspects not covered can be addressed.

9. A general revision of the article is suggested to improve the writing. Some typos (spaces and dots) are identified; for example, see lines 50, 52, 53, 70, 135, 174, 182, 189, 227, 228, 249, 257, 384, and 388.

Author Response

Point 1: It is recommended to review and better structure the title since it is not appropriate to use acronyms.

Response 1: According to expert’s opinion, we revised the title to “Load Identification for the More Electric Aircraft Distribution System Based on Intelligent Algorithm”.

Point 2: Check the definition of the acronym MEA on line 13, it should be: more electric aircraft (MEA). Besides in line 16: electrical power system (EPS).

Response 2: The corresponding parts are modified (highlighted in yellow).

Point 3: I recommend including an introductory paragraph in Sections 2, 3, 4, 4.2, and 5, indicating the topics described in these sections. This allows having a common thread at the time of reading.

Response 3: The corresponding parts are modified (highlighted in yellow).

Point 4: Present the flowchart of the genetic algorithm used.

Response 4: Please see the revised article for flowchart(Figure 5).

Point 5: In equation 10 use suitable parentheses to cover each fraction.

Response 5: We checked and corrected all formulas and their numbers in the text (highlighted in yellow).

Point 6: Please justify the parameter selection of the genetic algorithm.

Response 6: In section 4.3, we discussed different population size have affected the optimization ability of the algorithm. And we finally found that population size =30 is the best choice.

Besides, the values of ‘Generation of series’, ’Crossover rate’ and ‘Mutation rate’ are all selected from experience, and have been verified in the process of model training. The accuracy of the final identification results is 100%, which fully confirms the rationality of the values.

Point 6: Considering the stochastic behavior of the genetic algorithm, it is suggested to add a statistical analysis to strengthen the results regarding the methodology of parametric and non-parametric tests implemented to determine the comparison of cases (ANOVA, Kruskal Wallis, Bonferroni…). If the authors consider that these statistical tests are not feasible, it must be justified in the discussion section.

Response 7: Mean Squared Error (MSE) is often used as a performance measure for regression tasks. If MSE is employed to characterize identification accuracy, additional statistical analysis may be required. However, precision (PRE) and recall (REC) are suitable performance measures for classification models. Therefore, we adopted precision, recall and the harmonic average F1 to evaluate the accuracy of the identification results. Based on this, authors did not add a statistical analysis part.

We have added some words in Section 4.2 to express the rationale for choosing PRE, REC and F1 as a performance measures.

Point 8: It is important to include a discussion section where the effectiveness, limitations, and aspects not covered can be addressed.

Response 8: According to expert’s opinion, we have added corresponding statement in the final conclusions section 6 (highlighted in yellow).

Point 9: A general revision of the article is suggested to improve the writing. Some typos (spaces and dots) are identified; for example, see lines 50, 52, 53, 70, 135, 174, 182, 189, 227, 228, 249, 257, 384, and 388.

Response 9: We have checked and revived the whole text to improve the writing.

Reviewer 2 Report

In this article the authors are proposing a method to automatically identify the loads status on the remote power distribution unit of a more electric aircraft by using intelligent algorithm. 

The paper is well written, introduces a solid theoretical background, and offers good results and a thorough comparison to other approaches. I suggest to the authors to improve the english in the manuscript.

The paper is in a good shape for publication.

Author Response

Point 1: The paper is well written, introduces a solid theoretical background, and offers good results and a thorough comparison to other approaches. I suggest to the authors to improve the english in the manuscript.

Response 1: We have checked and revived the whole text to improve the writing. Thanks.
